# Gender Differences in Psychological Symptoms and Psychotherapeutic Processes in Japanese Children

**DOI:** 10.3390/ijerph17239113

**Published:** 2020-12-06

**Authors:** Toshio Kawai, Yuka Suzuki, Chihiro Hatanaka, Hisae Konakawa, Yasuhiro Tanaka, Aya Uchida

**Affiliations:** 1Kokoro Research Center, Kyoto University, Kyoto 606-8501, Japan; suzuki.yuka.4n@kyoto-u.ac.jp (Y.S.); konakawa.hisae.8e@kyoto-u.ac.jp (H.K.); 2Kokoro Research Center, Uehiro Uehiro Research Division, Kyoto University, Kyoto 606-8501, Japan; hatanaka.chihiro.3e@kyoto-u.ac.jp; 3Graduate School of Education, Kyoto University, Kyoto 606-8501, Japan; tanaka.yasuhiro.7u@kyoto-u.ac.jp; 4Graduate School of Human and Environmental Studies, Kyoto University, Kyoto 606-8501, Japan; ayauchi17@gmail.com

**Keywords:** gender differences, tic disorder, selective mutism, autism spectrum disorder, trichotillomania, aggression

## Abstract

Gender differences have been documented in the prevalence of psychological symptoms. Tic disorders and autism spectrum disorder (ASD) are more common in male clinical samples, while selective mutism and trichotillomania are more common in female clinical samples. In a review of 84 published case studies of Japanese children, this study explored gender differences in the prevalence of four categories of symptoms and expressions made in therapy for tics, selective mutism, autism spectrum disorder (ASD), and trichotillomania. Case studies were evaluated using both qualitative coding and statistical analysis. The findings were mostly consistent with epidemiological surveys and empirical research on adults. The gender differences in symptom prevalence and their expression could be summarized as differences in more direct aggression for boys versus indirect aggression for girls. The objective and progress in the therapy were to control impulsive energy for boys and to express energy for girls.

## 1. Introduction

There are gender differences in the prevalence of some psychological disorders, including within child clinical populations [1,2,3]. For example, epidemiological data show that selective mutism and trichotillomania tend to be more prevalent in girls, whereas tic disorder and autism spectrum disorder (ASD) tend to be more prevalent in boys [4,5]. Furthermore, gender differences can also be observed not only in diagnoses but in the particular symptoms or behaviors exhibited by children, even in cases with the same diagnosis. These gender differences are often consistent with general gender differences in healthy populations, suggesting clinical gender differences are found on a spectrum/dimension of behavior and may emerge in part due to more general socialization processes. For example, boys are more likely to exhibit externalizing behaviors, and girls are more likely to exhibit internalizing behaviors [6]. This tendency is also observed in terms of how children express aggression, whereby boys may externalize their aggression by taking physical action, often directed toward others, while girls may internalize their aggression by directing it inward toward the self or by showing more indirect relational aggression, such as sarcasm or passive aggression [7,8].

This gender difference in the tendency to internalize or externalize is one explanation for boys’ relatively higher diagnosis rate with attention deficit hyperactivity disorder (ADHD). It is estimated that in community samples, there are about four boys diagnosed for every girl [9,10]. One possible hypothesis is that boys are not necessarily more vulnerable to ADHD, but rather girls are more likely to have internalized rather than externalized symptoms. This then increases the likelihood of girls being overlooked at possible intervention points, such as when a student’s disruptive behavior gets the teacher’s attention. Likewise, socialization of gender, which may be more or less divergent depending on cultural contexts, could explain the tendency for males to be more inclined towards externalizing and females to internalizing behaviors [11,12]. Understanding the role of gender in not only the prevalence of disorders but also children’s expression of symptoms can contribute to more effective diagnosis and clinical treatment.

The current study selected four disorders with gender prevalence differences in past epidemiological research: tic disorder, selective mutism (or elective mutism), autism spectrum disorder (ASD, autism), and trichotillomania. Compared to ADHD and learning disabilities, these disorders are also theorized to have strong neurodevelopmental components [13] but remain less studied in terms of gender differences in symptom expression, particularly in Japan, where past approaches have primarily relied on the macro-level lens of epidemiological or population survey studies. Examining the symptom expressions of children with these disorders from the perspective of externalizing, direct aggression versus internalizing, and indirect aggression is particularly useful because these four disorders are similar in that they are all, in part, marked by the reduced ability to express one’s inner mental and emotional life through words. This is a developmentally difficult task for young children, but it is made more so by these disorders. Similarly, symptoms or characteristic expressions associated with these disorders, such as pulling one’s hair when anxious, reflect maladaptive ways that children may express themselves. As such, these disorders can be linked to disruptive behaviors that impair the child’s healthy development in the absence of effective interventions. While there are other disorders in childhood with similar qualities, such as nocturnal enuresis, we could not find enough case studies to include them in the review. Thus, this study examined whether the gender differences in the prevalence of these four disorders are also found in the expressions and behaviors of children in a clinical setting.

Sandplay therapy, in which children create scenes or landscapes in a sandbox with miniature toys, is recommended and especially effective for children experiencing difficulty in verbalizing self-insight and emotional expression [14]. Five of the authors are psychotherapists from a dynamic psychotherapy orientation with a particular focus on analytical psychology, which uses images, such as dreams and expressions by drawings in psychotherapy. Therefore, the theoretical framework and rationale for investigating and interpreting expressions and behaviors of children in sandplay therapy reflect the authors’ clinical expertise. The research team’s collective clinical observations of children during sandplay tended to find that boys were more likely to create scenes that involved fighting or wars—symbolic of externalization, whereas girls were more likely to create enclosed spaces in the sand, such as houses with rooms—symbolic of internalization [15]. If the distribution of case studies for a particular disorder was one girl to every four boys, we might also expect to observe similar differences in the ratio between particular internalizing, indirect expressions and externalizing, direct expressions.

### 1.1. Tic Disorder

Tics refer to compulsive, externalized behavioral tendencies, such as sudden twitches, movements, or verbalizations [16,17]. While tics are most commonly associated with Tourette syndrome, the DSM-5 (Diagnostic and Statistical Manual of Mental Disorders) and ICD-10 (International Statistical Classification of Diseases and Related Health Problems) also recognize that tics appear as symptoms of other diagnoses, including an unspecified tic disorder, among others [16,17]. Although there is some variability in the specific estimate in the literature, the overall tendency for tics is that it appears more prevalent among boys, such as in the case of Tourette’s syndrome, for which there are three-to-four boys diagnosed for every girl [18,19].

### 1.2. Selective Mutism

Selective mutism refers to a psychological condition whereby a child who is otherwise physically able and willing to speak to selected persons consistently refuses to speak in social situations [16,17]. Although there is variability in the estimated prevalence of selective mutism, some evidence indicates that girls may be more likely to exhibit selective mutism where the ratio of boys to girls is one boy to every 0.4–2.6 girls [20,21,22,23]. A recent review, including samples from Turkey, Sweden, Finland, and Ireland, also showed the ratio of males to females with selective mutism as approximately 1:1.5 [24].

### 1.3. Autism Spectrum Disorder (ASD)

Autism spectrum disorder is a neurodevelopmental disorder in which symptoms cause clinically significant impairment in social, occupational, or other important areas of functioning. According to the DSM-5, ASD refers to “persistent deficits in social communication and social interaction” and “restricted, repetitive patterns of behavior, interests, or activities” [16]. The estimated prevalence of ASD has been on the rise. For example, in 1979, autism was estimated to occur in 5 per 10,000 individuals, and ASD to occur in 21 per 10,000 individuals [25]. A more recent survey (2009) estimated ASD to occur in 157 per 10,000 individuals [26]. As to the gender difference, the mean male:female ratio shown in the above review paper was 4.2:1, which is consistent with work from this year in which the ratio of males to females was 4:1 [27], and consistent with research showing that boys are 4.3 times more likely to exhibit ASD symptoms than girls [28]. As for Japan, a cohort study showed that the ratio of males to females for autism was 2.5:1 [29].

### 1.4. Trichotillomania

In the DSM-5, trichotillomania is classified as a disorder related to obsessive-compulsive disorder. The diagnosis of trichotillomania considers the presence or absence of hair pulling, the extent of hair loss, and subsequent functional impairment, such as withdrawing from social outings due to worries over the hair’s appearance, while ruling out alternative causes, such as autoimmune disease, an atopic predisposition, or type 1 diabetes [30].

The empirical evidence regarding the magnitude of gender differences for trichotillomania is mixed. In clinical research, the ratio of males to females with trichotillomania is around 1:8 to 1:14 [31,32]. In one study examining over 2500 students, the prevalence was estimated at 1.5% of males and 3.4% of females [33]. However, this study included all cases with hair-pulling symptoms, even those cases that did not fully meet the DSM-5 criteria for trichotillomania. A clinical survey of 266 cases organized by the Japan Trichotillomania Improvement Association found females made up the majority of cases at 93.7% [34]. However, these clinical surveys were not limited to childhood trichotillomania. In contrast, epidemiological research shows a smaller gender gap, with about one boy for every two girls meeting the diagnostic criteria above. Finally, one epidemiological survey found no significant gender difference across 10,169 adults, with 1.8% of males and 1.7% of females having trichotillomania [35]. One possible interpretation for these varying estimates is that girls are more likely to present for psychotherapeutic services than boys, leading to overrepresentation in clinical estimates.

### 1.5. The Present Research

A review of the literature showed gender differences in child prevalence for these four disorder categories in epidemiological surveys (see Table 1 for a summary).

Most of the existing research on gender differences in psychopathology has been conducted using epidemiological and survey approaches and, often, taking place within Western cultures, such as the United States. While case studies may have small samples (usually N = 1) and may be subject to the influence of the clinicians’ perceptions, case studies act as useful complements and sources of converging evidence for these epidemiological and survey methods. Case studies include rich, detailed, in-depth information over time about clients in a clinical setting. The detailed information contained within case studies can also be useful for generating hypotheses or calibrating theoretical perspectives and may capture information overlooked in more structured or self-report studies. Thus, the current study’s aim was to examine whether converging evidence for gender prevalence differences could be obtained from a qualitative and quantitative review of 84 case studies spanning five decades and whether such differences in gender prevalence was also found when examining not only the diagnosis but the child’s expressions made in a clinical setting during sandplay therapy.

In summary, tic disorders and ASD appear to be more common in males, while selective mutism and trichotillomania appear to be more common in females. As a complement to the epidemiological and clinical survey research, we aimed to explore whether there are similar gender differences when examining clinical child case studies in Japan. We also hypothesized and explored whether children differ in how they express themselves in sandplay psychotherapy as a function of these disorders. Children who have externalizing behavioral problems tend to exhibit conflict through behaviors, such as aggression, hyperactivity, distraction, lying, and kleptomania [6,7,8]. Expressions (e.g., behaviors) that occurred in psychotherapy were examined as a function of disorder type and gender. We hypothesized that the direction of aggression is different according to disorders. In the case of tic disorder and ASD, in which more male prevalence is expected, we predicted that aggression would be displayed more directly (externalized), while in selective mutism and trichotillomania, in which more female prevalence is expected, we predicted that aggression would predominantly be directed inwardly and more restrained (internalized). Aggression was defined in this study as an intention and behavior to attack others or objects physically and/or psychologically. As such, we expected to see gender differences in the analyzed case studies, such that boys would display more externalizing behaviors and aggression, while girls would display more internalizing behaviors and aggression.

## 2. Materials and Methods

In order to explore gender differences in the prevalence of these four relatively less-studied disorders and to subsequently examine how children express themselves in psychotherapy, a team of five experienced psychotherapists analyzed 84 published child case studies. This was done by qualitatively coding the psychotherapeutic sessions by recording the expression of symptoms and turning points in the psychotherapeutic process for each case. This method of using qualitative evaluations and checklists by clinicians and coauthors is supported by previous research [36,37].

For the analysis of case studies, we selected three leading Japanese Clinical Psychology journals: The Journal of Japanese Clinical Psychology, which is a primary journal for clinical psychology in Japan, the Archives of Sandplay Therapy and the Japanese Journal of Play Therapy, both of which are leading child psychotherapy journals in Japan. Finally, to include cases prior to 1983, Kyoto University’s Annual Report of the Institute for Psychotherapy volumes 1 through 10 were included, resulting in cases from 1974 onward.

Within the journals, articles were selected by searching the titles and keywords for the following terms: “tic disorder”, “Tourette syndrome”, “mutism”, “selective mutism”, “ASD (autism spectrum disorder)”, “autism”, “Asperger”, and “trichotillomania”. Case studies of patients over 16 years old were not included as we focused on children in this research. Parent case studies and survey papers were also excluded from the analyzed sample. This resulted in a total of 84 cases—10 case studies for tic disorders, 21 cases for selective mutism, 48 cases for ASD, and 7 cases for trichotillomania (see Table 2). It should also be noted that the DSM-5’s current description of ASD is one iteration reflecting a number of earlier changes in both the criteria and approach to diagnosis for ASD [38]. Considering these changes, and because the case studies are taken from several decades, we included the DSM-5’s ASD diagnosis in addition to severe autism [39], the DSM-III’s Pervasive Developmental Disorder (1980), DSM-III-TR’s pervasive developmental disorder-not otherwise specified (1987), and three disorders in DSM-IV (1994), including Autistic Disorder, Asperger’s Disorder, and Pervasive Developmental Disorder—Not Otherwise Specified (PDD-NOS).

Each case was discussed by five experienced psychotherapists to identify (1) the characteristic psychological problems of the case and (2) the kind of progress or improvement in therapy. This research was approved by the Ethical Review Committee for Clinical Psychology Research at Kyoto University with the ethical approval code 20003.

## 3. Results

### 3.1. Qualitative Observations of the Case Studies

#### 3.1.1. Tic Disorder

While the search resulted in eight case studies for tic disorders, we found two ASD cases also contained tic symptomology; therefore, we included them in the analyses for tic disorders (and removed from ASD), resulting in ten tic disorder cases. The cases consisted of nine boys and one girl, whose ages ranged from 3 to 15 years (M = 8.30; SD = 3.07), revealing a gender difference that is consistent with existing research on tic disorders. As is typical for tic disorders, aggression was a common complaint in these case studies [40]. Boys were described as showing aggression more directly, as a type of externalizing behavior (such as hitting someone or something, biting the therapist, shouting abusive language) (total nine; 9:0 boys to girls). There was only one girl among the case studies, but the client was described as showing aggression more indirectly (such as using sarcastic tones while speaking, presenting cruel and grotesque stories and pictures). However, given the limited availability of girls with tic disorder case studies, we acknowledge it is not possible to draw strong conclusions about the generalizability of this case.

We found that over the course of therapy, children (total nine; 8:1 boys to girls) tended to increasingly verbally express their thoughts and feelings (e.g., a boy who shouted “silly” as a vocal tic later became able to verbally articulate criticisms in complete sentences). We also observed in all nine boys (9:0 boys to girls) a gradual increase in the ability to control impulses and aggression (e.g., a boy who could not initially control his shots in a ball game with the therapist and who would often use brute force, later made slower pitches and accepted time out). For some boys (total four; 4:0 boys to girls), we also recognized the emergence of the sense of differentiating between ‘good’ and ‘bad’ (e.g., a boy presented a story about police cars versus a criminal). Finally, in six boys (6:0 boys to girls), we observed that a ‘passage’ structure, such as a road or a circuit, was often built later in therapy, possibly symbolizing the unobstructed flow of emotions (e.g., a boy who drew a congested road at first, drew a road where cars drove smoothly in the later phases of therapy). We argue that these changes symbolize the ability of children with tic disorder to express their aggression in ways that are more socially acceptable.

#### 3.1.2. Selective Mutism

The 21 cases of selective mutism consisted of 5 boys and 16 girls whose ages ranged from 4 to 15 years (M = 9.29; SD = 3.21). More female cases aligned with epidemiological prevalence data. From our analysis, we found that boys tended to show more externalized behaviors with physical aggression, while girls (total 13; 2:11 boys to girls) expressed more internalized through indirect passive aggression (e.g., when a girl made a snake bite one of the important dolls in the story while playing). One unexpected observation was that children in selective mutism cases (total six; 0:6 boys to girls), but not other disorders, showed gender conflict (e.g., a girl said that she was a boy because her mother wanted a boy).

In seven cases (0:7 boys to girls), children placed all play items into a container or into a space that was contained by fences. We posit that this mirrors the inherent nature of selective mutism as a symptom in which inner psychological life exists but cannot be expressed. This is an extreme case of internalization. In 12 cases (2:10 boys to girls), children often used rare or unusual items and toys, such as using snakes and frogs in lieu of more everyday animals, such as dogs and cats. We argue this indicates hidden and repressed psychological contents of these children compounded by selective mutism. Consistent with these observations, psychotherapeutic turning points among selective mutism cases were often characterized by moments of discharge and expression (e.g., laughing or making loud sounds (e.g., using a horn)) (total 15; 2:13 boys to girls). Alternatively, therapeutic moments also occurred with more subtle self-expression, such as when children created stories or pictures, including self-portraits (total nine; 0:9 boys to girls). These stories or pictures often contained dual elements, where the child expressed or captured two layers of the world. For example, in a child’s story, there was another world “below ground”, which was different from the normal world, where both treasures and scary things simultaneously existed. We believe this suggests that the child developed self-consciousness and self-reflection regarding one’s inner mental life. In some cases, other evidence of therapeutic progress was observed toward the end of therapy, such as when the child would build a toy house and permit a figure to enter it (total 15; 2:13 boys to girls).

#### 3.1.3. ASD

There were 48 case studies for ASD, consisting of 39 boys and 9 girls, and their ages ranged from 1 to 15 years (M = 6.58; SD = 3.83). A gender difference was consistent with the epidemiological surveys listed in Table 1. The search result for ASD yielded the most case studies relative to the other selected symptoms, which aligns with the trend towards growing ASD prevalence. However, this increase may in part be driven by overgeneralization of ASD rather than an objective increase in its occurrence; specifically, in Japan, where psychiatrists and psychotherapists over-diagnose ASD [41,42]. This was reflected in the current review as there was a clinical consensus that five cases were wrongly diagnosed as ASD. The team concluded that two boys were more likely cases of tic disorders and, thus, were only analyzed with the tic disorders cases and that three case studies of girls were inconsistent with an ASD diagnosis and were omitted from all analyses. These three cases seemed to present an attachment disorder, anxiety disorder, and likely panic disorder, respectively. As a result, we analyzed 43 cases (37 boys; 6 girls) for ASD.

One interpretation of children with ASD is the difficulty in differentiating between self and others, as well as between existence and non-existence. Thus, in sandplay therapy, the therapeutic process often focuses on the establishment of these psychological boundaries and the ability to differentiate between the self and other, of existence and non-existence [43]. This was reflected in our observations of the majority of the case studies (total 26; 21:5 boys to girls), as separation was a recurring central theme (e.g., the separation between client and mother, between client and therapist, and between the client with items in the therapy room).

As for aggression, direct physical contact with the therapist (hitting or biting) was found in 16 (13:3 boys to girls) cases. This form of direct, physically externalized aggression can be interpreted as a primitive way of differentiating between the other and one’s self. Children with ASD (16:2 boys to girls) were also likely to enter into various small spaces, such as the sand tray or a box house, which likely provided a feeling of containment and embodiment (i.e., a secure base) [44]. Later in the psychotherapeutic process, some children (4:0 boys to girls) added toys to these contained spaces. For eight ASD cases (7:1 boys to girls), children used mirrors for self-gaze, suggesting further development of a sense of self, which is a precondition for differentiating between self and others [45]. In 12 cases, physical excretion or discharge (urination and defecating) occurred while in the playroom, perhaps as a physical means of self-expression [46]. Emptying the sandbox and using the number zero in the play happened in five cases (4:1 boys to girls), which may symbolize finding a starting base and point [47]. Over the course of therapy, 14 cases (13:1 boys to girls) sought out height, such as playing with tall items or climbing up to an elevated place, which may have to do with the emergence of an agentic subject and self. Thus, for ASD cases, the qualitative observations made by the clinical team showed variety in expressions. At the same time, they indicate that the externalizing tendency (physical contact with the therapist, physical discharge, and emphasis of height) is dominant in cases of ASD, which progresses toward an internalizing tendency (entering into small spaces and using mirrors) over the course of therapy.

#### 3.1.4. Trichotillomania

The seven cases of trichotillomania consisted of one boy and six girls, with ages ranging from 8 to 14 years (M = 11.29; SD = 1.83), again showing a gender ratio that is consistent with previous prevalence surveys. Similar to selective mutism, all cases of trichotillomania (with greater prevalence for girls) seemed to show that aggression was expressed indirectly (total seven; 1:6 boys to girls). However, in contrast to selective mutism, the expressions were simpler and employed fewer symbolic expressions. For example, in most cases (total five; 0:5 boys to girls), cohesive themes did not develop during therapy, and play mainly centered around the client’s nuanced and subtle aggression towards her mother (e.g., a girl forced her mother, who was very reluctant, to draw a picture and participate in a mutual picture-drawing game with her and her therapist). As a result of the relatively simpler therapeutic processes in trichotillomania, we found fewer expressions and qualitative interpretations for the analysis.

### 3.2. Quantitative Analysis of the Case Studies

To quantitatively evaluate the case studies, the team reviewed and scored all cases against a checklist. Based on the prominent themes from the qualitative analysis and in keeping with the distinction between externalization and internalization of behavior, the research team created a checklist of 23 categories of behaviors and symbolic imagery, which were regarded as characteristic and crucial for the therapeutic turning point for each disorder (Table 3). The 23 categories were determined from the 84 case studies and based partially on criteria from the DSM-5 and partially on the team of psychotherapist’s clinical expertise in sandplay and play therapy [14]. The objective of the checklist was to determine whether the qualitative differences reported above were statistically different.

Fisher’s exact tests were used to determine whether some psychotherapeutic symptoms/expressions were significantly more common in certain disorders. For statistically significant results, residual analyses identified which specific factor was driving statistical significance. Following recommendations by Cochran [48], we used Fisher’s exact tests instead of χ^2^-tests because more than 20% of cells had an expected value of less than five. Overall, gender differences in most of the expressions and therapeutic turning points derived from the qualitative analysis were supported by the quantitative analysis.

#### 3.2.1. Tic Disorder

Tic disorder cases were more likely to feature “quality of aggression (direct and simple)” (*p* < 0.01), “change of symptom to the verbal and humorous expression” (*p* < 0.01), “creation of interval and capacity to wait” (*p* < 0.01), “appearance of an image with a passage” (*p* < 0.01), “play demonstrating a differentiation between right and wrong” (*p* < 0.05), in comparison with selective mutism, ASD, and trichotillomania. These findings are consistent with the observations of characteristic expressions of tic disorder in the qualitative review of the case studies. While these characteristic expressions were also observed in other disorders, these expressions were significantly more common in tic disorder cases.

#### 3.2.2. Selective Mutism

Selective mutism cases were more likely to include “emergence of grotesque image and play” (*p* < 0.01), “change from controlled condition to laughter and making loud noise” (*p* < 0.01), “spontaneous appearance of symbolic story and picture” (*p* < 0.01), “emergence of self-portrait” (*p* < 0.01), “emergence of two layers of expression” (*p* < 0.01), “theme of gender conflict” (*p* < 0.01), “making a house and making a space for figures” (*p* < 0.01), and “quality of aggression (indirect and complicated)” (*p* < 0.01). Therefore, eight out of 23 characteristic expressions were significantly more common in selective mutism relative to tic disorder, ASD, and trichotillomania. This was also the highest number of characteristic expressions among the four symptom categories. However, we expected that “filling a container or fenced place with items” to be characteristic of selective mutism, but there was no statistically significant difference compared to other categories.

#### 3.2.3. ASD

As for ASD, “task (or exchange) of separation from mother or therapist” (*p* < 0.01), “client’s entering in the house box or sand tray” (*p* < 0.05), “the emergence of the image of height (for example, putting tall items, climbing up to high place)” (*p* < 0.05) were statistically more common relative to tic disorder, selective mutism, and ASD. Other expressions, which were regarded as typical for ASD in the qualitative review but were not found to be significantly more common in ASD, included “physical excretion and discharge in the playroom”, “direct physical contact with the therapist”, “emptying something”, “play looking into the mirror”.

#### 3.2.4. Trichotillomania

Trichotillomania showed significantly more “conflict with mother” (*p* < 0.01), “expression through images without symbolic development” (*p* < 0.01), “quality of aggression (indirect and complicated)” (*p* < 0.01) relative to tic disorder, selective mutism, and ASD. Similarly, these were identified as typical expressions of trichotillomania in the qualitative review.

#### 3.2.5. Gender Differences in Prevalence

To examine the relationship between gender and the symptoms/characteristic expressions in therapy, we aggregated all four symptom categories to test whether characteristic expressions differed by gender. χ^2^-test or Fisher’s exact test was used and was followed up with residual analyses. As in the case of analysis of symptoms, we used Fisher’s exact test instead of χ^2^-test following the suggestion by Cochran [48], if more than 20% of cells had an expected count of less than five. Table 4 shows the comparison of expressions in boys and girls.

Table 4 shows that there were significantly different gender differences in characteristic expressions and behaviors in psychotherapy. Relative to girls, the following expressions were more common in boys: “quality of aggression (direct and simple)” (*p* < 0.01), “creation of interval and capacity to wait” (*p* < 0.01), “task (or exchange) of separation from mother or therapist” (*p* < 0.05), “the emergence of the image of height (for example, putting tall items, climbing up to high place)” (*p* < 0.05). These same expressions were also found to be more common in tic disorder and ASD, which had a higher prevalence of case studies with boys.

In contrast, the following expressions were more common in girls compared to boys: “quality of aggression (indirect and complicated)” (*p* < 0.01), “filling a container or fenced place with items” (*p* < 0.01), “emergence of grotesque image and play” (*p* < 0.01), “change from controlled condition to laughter and making loud noise” (*p* < 0.01), “spontaneous appearance of symbolized story and picture” (*p* < 0.01), “emergence of self-portrait” (*p* < 0.01), “emergence of two layers of expression” (*p* < 0.01), “theme of gender conflict” (*p* < 0.01), “making a house and making a space for figures” (*p* < 0.01), “conflict with mother” (*p* < 0.01), “expression through images without symbolic development” (*p* < 0.05). These characteristics were also more common in selective mutism and trichotillomania, which had more case studies on girls.

## 4. Discussion

This research investigated the distribution of gender across case studies of tic disorders, selective mutism, ASD, and trichotillomania among Japanese children and aimed to determine whether the psychotherapeutic expressions characteristic of each disorder also differed by gender. We found that there were not only gender differences in the distribution of case studies based on the disorder diagnosis, but there were also gender-specific expressions of these disorders in psychotherapy. The same psychotherapeutic expressions that characterized tic disorder and ASD were also more common in boys than in girls. Meanwhile, the psychotherapeutic expressions that characterized selective mutism and trichotillomania were more common in girls compared to boys.

The higher prevalence of selective mutism and trichotillomania in girls was accompanied by more complicated and indirect expressions of aggression. Meanwhile, the higher prevalence of tic disorder in boys was associated with more direct and externalized expressions of aggression. In the case of tic disorder and selective mutism, these expressions also tended to align with the symptoms of the disorder and the subsequent objectives of psychotherapy. For example, one treatment goal for tic disorders is to control the impulsive energy, and one treatment goal for selective mutism is to express hidden or held-back energy. Therefore, more direct aggression at the initial phase of psychotherapy for tic disorder may reflect the need to control impulsive energy. While the indirect aggression in selective mutism reflects the existence of hidden energies and the expression of it.

Children with selective mutism appeared to have more differentiated and self-reflective mental structures as if they recognized two layers to their psychological world, in which the inward-focused, self-reflective function had become most dominant. This was in contrast to children with ASD, where the primary issue appeared to be building basic psychological structures, including the sense of self and differentiation between self and others. This contrast in clinical samples mirrors general gender differences in healthy, typical samples, in which girls tend to have a tendency to internalize, which may contribute to more differentiated psychological structures (i.e., differentiation of the internal from the external world).

However, there are several limitations to the current study, which should be kept in mind when drawing conclusions. First, like most published research, this study relied on published case studies for its review. It is possible that unpublished case studies introduce the “file-drawer problem”, in which there is some qualitative difference in the content of unpublished versus published case studies as it relates to these four disorders and gender. Although this limitation cannot be ruled out, the current analyses did generally converge with other forms of research that employ large-sample approaches. Second, the checklist created by the research team for this study was not externally created, as it was created based on the case studies that were analyzed. However, we aimed to minimize bias by independently reviewing each case study to inform the checklist categories. The checklist was created through clinical expertise and the current study’s focus on differentiating forms of indirect/internalizing and direct/externalizing forms of aggressive expression between genders in a sandplay clinical setting. Therefore, it is possible that other expressions relevant to this particular research question were not captured by the checklist. Future research could consider the psychometric assessment of the current checklist and compare whether different research teams generate similar items.

Finally, it remains an open question whether the gender differences for these disorders are bound to historical and cultural parameters. This study represents an emic approach, as the case studies are about Japanese children, reported by Japanese clinicians, and evaluated by a Japanese research team. While prevalence and symptomology largely align with epidemiological and clinical work from non-Japanese sources, we acknowledge the impact that cultural context has on informing symptom manifestations and the diagnostic process [49,50]. For example, Japan is known to have the culture-bound syndromes, Hikikomori (a refusal to leave the bedroom [51]), which is a manifestation of social and environmental forces prevalent within the Japanese cultural context [52,53]. Likewise, even for analogous syndromes, such as depression, symptomology can differ by culture, driven in part by cultural norms [54]. More research is needed to address whether analogous prevalence in symptomology is the result of analogous social and cultural forces. Similarly, psychotherapists and clients from the same cultural context not only share and have insight into the same socio-environment but are also trained within this specific environment. While this could engender culturally-specific skills in recognizing, interpreting, and relaying the meaning of certain expressions in psychotherapy, we argue this provides rich insight into a broader understanding of human psychology by contributing an analysis from the perspective of the Japanese cultural context.

Furthermore, our review of case studies is based on cases in the last 50 years. However, it is important to study the change of psychological consciousness and disorders over time. For example, the increasing prevalence of ASD in recent years could have to do with the change of dominant consciousness, from a modern, reflective consciousness to the so-called ‘postmodern consciousness’—characterized by dissociation and multiple selves [55,56]. The growing acceptance of sexual minorities in mainstream culture is another cultural change over time that may influence the understanding of gender, which, in turn, can have implications for gender differences in prevalence and in gender-specific development. Further study is needed to explore the change of disorders and psychotherapy over large timeframes, as this would also have ramifications for diagnosis and treatment effectiveness.

## 5. Conclusions

In spite of these limitations, the current research presented a unique combination of qualitative and quantitative analysis on expressions that occur during therapy while employing a culturally emic approach to case studies spanning five decades in Japan. Informed by the authors’ expert clinical observations of Japanese children during sandplay and play therapy, this exploratory study found that expressions and behaviors in psychotherapy are likely related to gender differences in the prevalence of specific disorders. While boys are more likely to express themselves in therapy with direct and externalized aggression, girls are more likely to express themselves with indirect, internalized aggression, and these tendencies in boys and girls align with the prevalence of which disorders they are likely to be diagnosed with. In this sense, therapeutic goals may differ by gender, in which boys work on differentiation and control of aggression, and girls work on the discharge of emotions and energy outwardly. The current research, therefore, explored gender-specific behaviors of disorders, in addition to gender-specific psychological development in therapy. Moreover, the therapeutic goals of integrating the characteristics associated with other genders—for boys, internalization, and for girls, externalization—can be an interesting suggestion for the topic of gender.

## Figures and Tables

**Table 1 ijerph-17-09113-t001:** Summary of prevalence and gender differences in existing epidemiological resources.

Symptom	Prevalence	Gender Difference	Source
Tourette’s syndrome	0.008%	2–4 boys: 1 girl	DSM-5 [16]
Tourette’s syndrome	0.003%	2 boys: 1 girl	ICD-10 [17]
Tourette’s syndrome	0.3%	3 boys: 1girl	2009 U.S. survey by Scahill, L. B. R. H et al. [18]
Tourette’s syndrome	-	3–4 boys: 1 girl	review by Eapen, V. et al. [19]
Selective Mutism	0.03–1%	More commonly observed in girls	ICD-10 [17]
Selective Mutism	0.03–1.89%	1 boy: 0.4–1.5 girls	review by Cho and Sonoyama [20]
Selective Mutism	-	More commonly observed in girls	survey by Kumpulainen, K. [21]
Selective Mutism	-	More commonly observed in girls	survey by Standart and Le Couteur, A. [22]
Selective (Elective) Mutism	-	1 boy: 1.5–2.6 girls	U.S. survey by Steinhausen. H.C. and Juzi, C. [23]
Selective Mutism	-	1 boy: 1.5 girls	survey by Hartung. [24]
Autism Spectrum Disorder	-	4 boys: 1 girl	survey by Azeez, Azeezat. [27]
Autism Spectrum Disorder	0.019%	4.3 boys: 1girl	U.S. Survey in 2020 by Maenner, M.J. et al. [28]
Autism	0.27%	2.5 boys: 1girl	Japan Servey in 2005 by Honda, H. et al. [29]
Trichotillomania	-	1 boy: 8–10 girls	survey by Panza, K. et al. [31]
Trichotillomania	-	1 boy: 14 girls	survey by Bottesi, G. et al. [32]

**Table 2 ijerph-17-09113-t002:** Summary of case studies included in the review.

No.	Reference	Clinician Sex	Client Sex	Age	Time of Treatment	Symptom	Journal	Notes
1	Suga, 1974	Female	Male	7	17 sessions	Tic disorders	Kyoto University’s Annual Report of the Institute for Psychotherapy	This case was analyzed as a tic disorder.
2	Swai, 1975	Female	Female	7	6 months, 23 sessions	Tic disorders	Kyoto University’s Annual Report of the Institute for Psychotherapy	This case was analyzed as a tic disorder.
3	Moritani, 1976	Male	Male	8	1 year 10 months, 56 sessions	Tic disorders, Gilles de la tourette’s syndrome	Kyoto University’s Annual Report of the Institute for Psychotherapy	This case was analyzed as a tic disorder.
4	Kuramitsu, 1979	Male	Male	15	1 year, 37 sessions	Tic disorders, Hyperactivity	Kyoto University’s Annual Report of the Institute for Psychotherapy	This case was analyzed as a tic disorder.
5	Kobayashi, 1999	Female	Male	7	1 year 3 months, 35 sessions	Gilles de la tourette’s syndrome	Journal pf Japanese Clinical Psychology	This case was analyzed as a tic disorder.
6	Kawasaki, 2000	Male	Male	9	63 sessions	Tic disorders	Journal pf Japanese Clinical Psychology	This case was analyzed as a tic disorder.
7	Umemura, 2011	Male	Male	10	3 years 8 months	Tic disorders, Autonomic dystonia, Irritable bowel syndrome	Journal pf Japanese Clinical Psychology	This case was analyzed as a tic disorder.
8	Harada, 2012	Male	Male	6	2 years 9 months, 86 sessions	Tic disorders	Archives of Sandplay Therapy	This case was analyzed as a tic disorder.
9	Fuji, 1976	Female	Female	9	8 sessions	Elective mutism, Dysarthria	Kyoto University’s Annual Report of the Institute for Psychotherapy	This case was analyzed as a selective mutism.
10	Kawata, 1977	Female	Female	4	1 year 2 months, 32 sessions	Elective mutism	Kyoto University’s Annual Report of the Institute for Psychotherapy	This case was analyzed as a selective mutism.
11	Ishi, 1978	Female	Female	12	1 year, 27 sessions	Elective mutism	Kyoto University’s Annual Report of the Institute for Psychotherapy	This case was analyzed as a selective mutism.
12	Tsuji, 1979	Female	Female	5	9 months, 20 sessions	Elective mutism	Kyoto University’s Annual Report of the Institute for Psychotherapy	This case was analyzed as a selective mutism.
13	Moriya, 1979	Female	Female	6	8 months, 22 sessions	Elective mutism	Kyoto University’s Annual Report of the Institute for Psychotherapy	This case was analyzed as a selective mutism.
14	Yurimoto, 1981	Female	Female	8	1 year 4 months, 30 sessions	Elective mutism, Night urine	Kyoto University’s Annual Report of the Institute for Psychotherapy	This case was analyzed as a selective mutism.
15	Kobayashi, 1982	Male	Male	11	2 years, 55 sessions	Elective mutism	Kyoto University’s Annual Report of the Institute for Psychotherapy	This case was analyzed as a selective mutism.
16	Moriya, 1983	Female	Female	8	1 year 5 months, 49 sessions	Elective mutism, Night urine	Kyoto University’s Annual Report of the Institute for Psychotherapy	This case was analyzed as a selective mutism.
17	Yamaguchi, 1983	Female	Male	13	1 year, 47 sessions	Elective mutism	Kyoto University’s Annual Report of the Institute for Psychotherapy	This case was analyzed as a selective mutism.
18	Ito, 1983	Female	Female	8	5 months, 14 sessions	Elective mutism	Kyoto University’s Annual Report of the Institute for Psychotherapy	This case was analyzed as a selective mutism.
19	Toyoda, 1983	Female	Female	7	1 year, 35 sessions	Elective mutism	Kyoto University’s Annual Report of the Institute for Psychotherapy	This case was analyzed as a selective mutism.
20	Ogawa, 1996	Female	Female	9	1 year 6 months, 29 sessions	Elective mutism	Journal pf Japanese Clinical Psychology	This case was analyzed as a selective mutism.
21	Otsuka, 1997	Female	Female	13	3 years	Elective mutism	Journal pf Japanese Clinical Psychology	This case was analyzed as a selective mutism.
22	Yasunaga & Tomonaga, 2004	Female	Female	9	1 years 6 months, 28 sessions	Elective mutism	The Japanese Journal of Play Therapy	This case was analyzed as a selective mutism.
23	Takashima, 2007	Male	Male	10	3 years, 121 sessions	Selective mutism	Journal pf Japanese Clinical Psychology	This case was analyzed as a selective mutism.
24	Ueno, 2010	Female	Male	8	1 year 8 months, 42 sessions	Selective mutism	Journal pf Japanese Clinical Psychology	This case was analyzed as a selective mutism.
25	Oda, 2010	Female	Female	4	2 years 3 months, 47 sessions	Elective mutism	The Japanese Journal of Play Therapy	This case was analyzed as a selective mutism.
26	Maruyama, 2013	Male	Male	13	3 years, 36 sessions	Elective mutism	Journal pf Japanese Clinical Psychology	This case was analyzed as a selective mutism.
27	Kuwabara, 2015	Female	Female	15	2 years 5months, 58 sessions	Selective mutism	Archives of Sandplay Therapy	This case was analyzed as a selective mutism.
28	Kakuda, 2016	Male	Female	15	8 months, 25 sessions	Selective mutism, Mild developmental disorders	Archives of Sandplay Therapy	This case was analyzed as a selective mutism.
29	Sunami, 2018	Female	Female	8	6years, 15 sessions	Selective mutism	The Japanese Journal of Play Therapy	This case was analyzed as a selective mutism.
30	Harima, 1974	Female	Male	3	1 year 2 months	Autism	Kyoto University’s Annual Report of the Institute for Psychotherapy	This case was analyzed as a autism spectrum disorder.
31	Kondo, 1975	Female	Male	3	1 year 6 months	Autism	Kyoto University’s Annual Report of the Institute for Psychotherapy	This case was analyzed as a autism spectrum disorder.
32	Hayashi, 1975	Female	Male	6	1 year	Autistic tendency	Kyoto University’s Annual Report of the Institute for Psychotherapy	This case was analyzed as a autism spectrum disorder.
33	Ueda, 1976	Female	Female	3	6 months	Autisic tendency, Mild intellectual disability, Attachment disorder	Kyoto University’s Annual Report of the Institute for Psychotherapy	This case was analyzed not be consistent with an ASD diagnosis and were omitted from all analyses.
34	Ishi, 1976	Female	Male	4	11 months	Autistic tendency	Kyoto University’s Annual Report of the Institute for Psychotherapy	This case was analyzed as a autism spectrum disorder.
35	Oi, 1976	Male	Male	4	1 year 1 months	Autism	Kyoto University’s Annual Report of the Institute for Psychotherapy	This case was analyzed as a autism spectrum disorder.
36	Kondo, 1976	Female	Male	2	6 years	Autisic tendency, Mild intellectual disability	Kyoto University’s Annual Report of the Institute for Psychotherapy	This case was analyzed as a autism spectrum disorder.
37	Sueta, 1976	Male	Male	3	1 year 4 months	Autism	Kyoto University’s Annual Report of the Institute for Psychotherapy	This case was analyzed as a autism spectrum disorder.
38	Soma, 1976	Male	Male	5	2 years 2 months	Autisic tendency, Mild intellectual disability	Kyoto University’s Annual Report of the Institute for Psychotherapy	This case was analyzed as a autism spectrum disorder.
39	Kuramitsu, 1977	Male	Male	3	9 months	Autism	Kyoto University’s Annual Report of the Institute for Psychotherapy	This case was analyzed as a autism spectrum disorder.
40	Kanno, 1978	Male	Male	6	1 year 2 months	Autistic tendency	Kyoto University’s Annual Report of the Institute for Psychotherapy	This case was analyzed as a autism spectrum disorder.
41	Inoue, 1982	Male	Male	8	1 year 1 month	Asperger syndrome	Kyoto University’s Annual Report of the Institute for Psychotherapy	This case was analyzed as a autism spectrum disorder.
42	Horiguchi, 1979	Female	Female	5	9 months	Autistic tendency	Kyoto University’s Annual Report of the Institute for Psychotherapy	This case was analyzed as a autism spectrum disorder.
43	Matsubara, 1980	Female	Female	4	1 year 6 months	Autisic tendency, Mild intellectual disability	Kyoto University’s Annual Report of the Institute for Psychotherapy	This case was analyzed as a autism spectrum disorder.
44	Kimura, 1981	Female	Male	6	11 months	Develomental disorder	Kyoto University’s Annual Report of the Institute for Psychotherapy	This case was analyzed as a autism spectrum disorder.
45	Yoshimura, 1982	Female	Male	3	2 years 9 months	Autistic tendency	Kyoto University’s Annual Report of the Institute for Psychotherapy	This case was analyzed as a autism spectrum disorder.
46	Kimura, 1983	Female	Male	3	1 year 5 months	Autistic tendency, Tic disorders	Kyoto University’s Annual Report of the Institute for Psychotherapy	This case was analyzed with tic disorders and not ASD.So we included in the analyses for tic disorders (and removed from ASD).
47	Kaito, 1983	Male	Female	7	8 months	Autistic tendency	Kyoto University’s Annual Report of the Institute for Psychotherapy	This case was analyzed as a autism spectrum disorder.
48	Sakaki & Imagawa, 1997	Male	Male	11	2 years 3 months, 49 sessions	Pervasive developmental disorder, Tic disorders	Journal pf Japanese Clinical Psychology	This case was analyzed with tic disorders and not ASD.So we included in the analyses for tic disorders (and removed from ASD).
49	Hirai, 1997	Male	Male	3	1 year 9 months, 250 sessions	Autism	Journal pf Japanese Clinical Psychology	This case was analyzed as a autism spectrum disorder.
50	Tsujikawa, 1999	Male	Male	10	1 year 11 months, 33 sessions	Autistic tendency	Journal pf Japanese Clinical Psychology	This case was analyzed as a autism spectrum disorder.
51	Ishikawa, 2005	Male	Male	8	2 years 8 months, 79 sessions	Autism	Journal pf Japanese Clinical Psychology	This case was analyzed as a autism spectrum disorder.
52	Furuichi, 2008	Female	Male	1	1 year 7 months, 54 sessions	Autism	Journal pf Japanese Clinical Psychology	This case was analyzed as a autism spectrum disorder.
53	Sato, 2010	Male	Male	14	1 year 11 months, 62 sessions	Autistic tendency	Journal pf Japanese Clinical Psychology	This case was analyzed as a autism spectrum disorder.
54	Takenaka, 2010	Female	Male	3	1 year, 29 sessions	Pervasive developmental disorder	Journal pf Japanese Clinical Psychology	This case was analyzed as a autism spectrum disorder.
55	Sakanaka, 2010	Male	Male	14	1 year 9 months, 58 sessions	Autism	Journal pf Japanese Clinical Psychology	This case was analyzed as a autism spectrum disorder.
56	Siomoto, 2011	Male	Male	15	1 year, 31 sessions	Pervasive developmental disorder	Journal pf Japanese Clinical Psychology	This case was analyzed as a autism spectrum disorder.
57	Yoshioka & Furuta, 2011	Male	Male	5	1 year 6 months, 50 sessions	Pervasive developmental disorder	Archives of Sandplay Therapy	This case was analyzed as a autism spectrum disorder.
58	Kamiya, 2011	Male	Male	13	2 years 11 months, 93 sessions	Develomental disorder	Archives of Sandplay Therapy	This case was analyzed as a autism spectrum disorder.
59	Tanaka, 2012	Male	Male	6	2 years, 77 sessions	Autism	Archives of Sandplay Therapy	This case was analyzed as a autism spectrum disorder.
60	Hamada, 2012	Female	Male	5	2 years 8 months, 57 sessions	Autistic tendency	The Japanese Journal of Play Therapy	This case was analyzed as a autism spectrum disorder.
61	Koyama, 2013	Male	Male	9	29 sessions	Asperger syndrome	Journal pf Japanese Clinical Psychology	This case was analyzed as a autism spectrum disorder.
62	Furuichi, 2014	Female	Male	10	4 years 5 months, 38 sessions	Develomental disorder	Archives of Sandplay Therapy	This case was analyzed as a autism spectrum disorder.
63	Ozawa, 2014	Male	Male	7	5 years, 108 sessions	Autisic tendency, Mild intellectual disability	Journal pf Japanese Clinical Psychology	This case was analyzed as a autism spectrum disorder.
64	Masumi & Harizuka, 2014	Female	Male	12	1 year, 44 sessions	Autism	Journal pf Japanese Clinical Psychology	This case was analyzed as a autism spectrum disorder.
65	Towhata, 2014	Male	Male	12	1 year, 39 sessions	Pervasive developmental disorder	Archives of Sandplay Therapy	This case was analyzed as a autism spectrum disorder.
66	Sugiyama, 2014	Female	Female	6	5 months,17 sessions	Autism	Archives of Sandplay Therapy	This case was analyzed as a autism spectrum disorder.
67	Sugiyama, 2016	Female	Female	5	7 years 6 months, 180 sessions	High-functioning autism	Archives of Sandplay Therapy	This case was analyzed as a autism spectrum disorder.
68	Hasegawa, 2016	Female	Male	4	4 years, 95 sessions	Autistic tendency	Archives of Sandplay Therapy	This case was analyzed as a autism spectrum disorder.
69	Fujimaki, 2016	Female	Male	3	6 months, 9 sessions	Autism spectrum disorder	Archives of Sandplay Therapy	This case was analyzed as a autism spectrum disorder.
70	Hoffman, 2016	Male	Male	4	3 years 4 months, 102 sessions	Autism	Archives of Sandplay Therapy	This case was analyzed as a autism spectrum disorder.
71	Tadokoro, 2016	Female	Female	8	2 years 5 months, 21 sessions	Anxiety disorder, Pervasive developmental disorder	The Japanese Journal of Play Therapy	This case was analyzed not be consistent with an ASD diagnosis and were omitted from all analyses.
72	Nagira, 2017	Female	Female	15	3 years 3 months	Asperger syndrome, Anxiety disorder	Journal pf Japanese Clinical Psychology	This case was analyzed not be consistent with an ASD diagnosis and were omitted from all analyses.
73	Umemura, Hasegawa, Hatanaka & Tanaka, 2017	Male, Female	Male	2	3 years 2 months, 121 sessions	Autism spectrum disorder	Archives of Sandplay Therapy	This case was analyzed as a autism spectrum disorder.
74	Fujimaki, 2017	Female	Male	9	7 years, 156 sessions	Autism spectrum disorder	Archives of Sandplay Therapy	This case was analyzed as a autism spectrum disorder.
75	Miyata, 2017	Female	Male	5	13 years, 209 sessions	Autism spectrum disorder	The Japanese Journal of Play Therapy	This case was analyzed as a autism spectrum disorder.
76	Noguchi, 2018	Male	Male	14	1 year 7 months, 55 sessions	Autistic tendency, Obsessive-compulsive symptoms	Archives of Sandplay Therapy	This case was analyzed as a autism spectrum disorder.
77	Sakamoto, 2019	Female	Female	12	1 year, 38 sessions	Autistic tendency	Archives of Sandplay Therapy	This case was analyzed as a autism spectrum disorder.
78	Watanabe, 2000	Male	Male	8	2 years 6 months, 67 sessions	Trichotillomania	Journal pf Japanese Clinical Psychology	This case was analyzed as a trichotillomania.
79	Miyato & Inoue, 2004	Female	Female	10	6 years 7 months, 257 sessions	Trichotillomania	Journal pf Japanese Clinical Psychology	This case was analyzed as a trichotillomania.
80	Ikeda, 2005	Male	Female	14	2 years, 11sessions	Tricobezor	Journal pf Japanese Clinical Psychology	This case was analyzed as a trichotillomania.
81	Kiyotaki, 2006	Female	Female	11	15 sessions	Trichotillomania	The Japanese Journal of Play Therapy	This case was analyzed as a trichotillomania.
82	Miyazaki, 2013	Male	Female	13	1 year, 18 sessions	Trichotillomania	Archives of Sandplay Therapy	This case was analyzed as a trichotillomania.
83	Ishikawa, 2018	Male	Female	11	1 year 4 months. 18 sessions	Developmental trauma disorder, Trichotillomania	Archives of Sandplay Therapy	This case was analyzed as a trichotillomania.
84	Komai, 2018	Female	Female	12	10 months, 9 sessions	Trichotillomania	The Japanese Journal of Play Therapy	This case was analyzed as a trichotillomania.

**Table 3 ijerph-17-09113-t003:** The appearance of characteristic expressions and changes in psychotherapy.

Characteristic Expressions	Symptoms										
	Tic Disorder		Mutism		ASD		Trichotillomania		χ^2^	*p* Value	Effect Size (Cramer’V)
	Yes	No	Yes	No	Yes	No	Yes	No			
Quality of aggression (direct and simple)	9 (3.6)	1 (−3.6)	5 (−1.6)	16 (1.6)	17 (0.2)	26 (−0.2)	0 (−2.2)	7 (2.2)	Fisher’s exact test	*p* < 0.01	0.47
Quality of aggression (indirect and complicated)	1 (−1.9)	9 (1.9)	13 (2.7)	8 (−2.7)	9 (−3.2)	34 (3.2)	7 (3.6)	0 (−3.6)	Fisher’s exact test	*p* < 0.01	0.56
Change of symptom to verbal and humorous expression	9 (6.8)	1 (−6.8)	3 (−0.3)	18 (0.3)	0 (−4.2)	43 (4.2)	1 (−0.1)	6 (0.1)	Fisher’s exact test	*p* < 0.01	0.78
Creation of interval and capacity to wait	9 (7.1)	1 (−7.1)	0 (−2.2)	21 (2.2)	3 (−2.1)	40 (2.1)	0 (−1.2)	7 (1.2)	Fisher’s exact test	*p* < 0.01	0.80
Play demonstrating a differentiation between right and wrong	4 (3.1)	6 (−3.1)	1 (−1.1)	20 (1.1)	4 (−0.6)	39 (0.6)	0 (−1.0)	7 (1.0)	Fisher’s exact test	*p* < 0.05	0.36
Appearance of an image with a passage	6 (4.6)	4 (−4.6)	2 (−0.6)	19 (0.6)	3 (−1.8)	40 (1.8)	0 (−1.1)	7 (1.1)	Fisher’s exact test	*p* < 0.01	0.51
Filling a container or fenced place with items	1 (−0.6)	9 (0.6)	7 (2.5)	14 (−2.5)	4 (−1.8)	39 (1.8)	1 (−0.1)	6 (0.1)	Fisher’s exact test	n.s.	0.28
Emergence of grotesque image and play	2 (0.0)	8 (0.0)	12 (5.0)	9 (−5.0)	2 (−3.6)	41 (3.6)	0 (−1.4)	7 (1.4)	Fisher’s exact test	*p* < 0.01	0.57
Change from controlled condition to laughter and making loud noise	1 (−1.3)	9 (1.3)	15 (5.3)	6 (−5.3)	6 (−2.8)	37 (2.8)	0 (−1.7)	7 (1.7)	Fisher’s exact test	*p* < 0.01	0.60
Spontaneous appearance of symbolized story and picture	3 (−0.6)	7 (0.6)	15 (3.6)	6 (−3.6)	11 (−2.5)	32 (2.5)	2 (−0.6)	5 (0.6)	Fisher’s exact test	*p* < 0.01	0.41
Emergence of self-portrait	0 (−1.3)	10 (1.3)	9 (4.9)	12 (−4.9)	1 (−2.9)	42 (2.9)	0 (−1.0)	7 (1.0)	Fisher’s exact test	*p* < 0.01	0.55
Emergence of two layers of expression	0 (−1.6)	10 (1.6)	13 (5.9)	8 (−5.9)	2 (−3.4)	41 (3.4)	0 (−1.3)	7 (1.3)	Fisher’s exact test	*p* < 0.01	0.66
Theme of gender conflict	0(−1.0)	10(1.0)	6(3.8)	15(−3.8)	1(−2.2)	42(2.2)	0(−0.9)	7(0.9)	Fisher’s exact test	*p* < 0.01	0.42
Making a house and making a space for figures	2 (−0.4)	8 (0.4)	15 (5.8)	6 (−5.8)	3 (−3.9)	40 (3.9)	0 (−1.6)	7 (1.6)	Fisher’s exact test	*p* < 0.01	0.65
Task (or exchange) of separation from mother or therapist	3 (−0.5)	7 (0.5)	0 (−4.1)	21 (4.1)	26 (4.6)	17 (−4.6)	1 (−1.3)	6 (1.3)	Fisher’s exact test	*p* < 0.01	0.55
Physical excretion and discharge in the play room	1 (−0.8)	9 (0.8)	3 (−0.7)	18 (0.7)	12 (2.0)	31 (−2.0)	0 (−1.4)	7 (1.4)	Fisher’s exact test	n.s.	0.24
Direct physical contact with the therapist	1 (−1.2)	9 (1.2)	2 (−1.9)	19 (1.9)	16 (2.8)	27 (−2.8)	1 (−0.7)	6 (0.7)	Fisher’s exact test	n.s.	0.31
Client’s entering in the house box or sand tray	1 (−1.4)	9 (1.4)	4 (−1.1)	17 (1.1)	18 (2.9)	25 (−2.9)	0 (−1.7)	7 (1.7)	Fisher’s exact test	*p* < 0.05	0.34
Emptying something	1 (0.3)	9 (−0.3)	0 (−1.5)	21 (1.5)	5 (1.5)	38 (−1.5)	0 (−0.8)	7 (0.8)	Fisher’s exact test	n.s.	0.21
Play looking into the mirror	0 (−1.2)	10 (1.2)	1 (−1.1)	20 (1.1)	8 (2.3)	35 (−2.3)	0 (−1.0)	7 (1.0)	Fisher’s exact test	n.s.	0.26
Emergence of the image of height	4 (1.3)	6 (−1.3)	1 (−2.3)	20 (2.3)	14 (2.1)	29 (−2.1)	0 (−1.5)	7 (1.5)	Fisher’s exact test	*p* < 0.05	0.35
Conflict with mother	1 (0.0)	9 (0.0)	2 (−0.1)	19 (0.1)	0 (−3.2)	43 (3.2)	5 (5.7)	2 (−5.7)	Fisher’s exact test	*p* < 0.01	0.65
Expression through images without symbolic development	0 (−0.8)	10 (0.8)	0 (−1.2)	21 (1.2)	0 (−2.2)	43 (2.2)	4 (6.7)	3 (−6.7)	Fisher’s exact test	*p* < 0.01	0.74

Note. Adjusted standardized residuals appear in parentheses below group frequencies. The shaded cells indicate statistical significance (defined as *p* < 0.05).

**Table 4 ijerph-17-09113-t004:** The appearance of these characteristic expressions and gender difference.

Characteristic Expressions	Gender Differences						
	Males		Females		χ^2^	*p* Value	Effect Size (φ)
	Yes	No	Yes	No			
Quality of aggression (direct and simple)	26 (2.7)	30 (−2.7)	5 (−2.7)	25 (2.7)	7.51	*p* < 0.01	0.30
Quality of aggression (indirect and complicated)	11 (−4.1)	45 (4.1)	19 (4.1)	11 (−4.1)	16.42	*p* < 0.01	0.44
Change of symptom to verbal and humorous expression	8 (−0.3)	48 (0.3)	5 (0.3)	25 (−0.3)	Fisher’s exact test	n.s.	0.32
Creation of interval and capacity to wait	12 (2.7)	44 (−2.7)	0 (−2.7)	30 (2.7)	Fisher’s exact test	*p* < 0.01	0.30
Play demonstrating a differentiation between right and wrong	8 (1.6)	48 (−1.6)	1 (−1.6)	29 (1.6)	Fisher’s exact test	n.s.	0.17
Appearance of an image with a passage	9 (1.2)	47 (−1.2)	2 (−1.2)	28 (1.2)	Fisher’s exact test	n.s.	0.13
Filling a container or fenced place with items	4 (−2.8)	52 (2.8)	9 (2.8)	21 (−2.8)	Fisher’s exact test	*p* < 0.01	0.30
Emergence of grotesque image and play	4 (−3.7)	52 (3.7)	12 (3.7)	18 (−3.7)	Fisher’s exact test	*p* < 0.01	0.40
Change from controlled condition to laughter and making loud noise	8 (−3.3)	48 (3.3)	14 (3.3)	16 (−3.3)	10.76	*p* < 0.01	0.35
Spontaneous appearance of symbolized story and picture	12 (−3.9)	44 (3.9)	19 (3.9)	11 (−3.9)	14.88	*p* < 0.01	0.42
Emergence of self−portrait	0 (−4.6)	56 (4.6)	10 (4.6)	20 (−4.6)	Fisher’s exact test	*p* < 0.01	0.50
Emergence of two layers of expression	2 (−4.6)	54 (4.6)	13 (4.6)	17 (−4.6)	21.45	*p* < 0.01	0.50
Theme of gender conflict	0 (−3.8)	56 (3.8)	7 (3.8)	23 (−3.8)	Fisher’s exact test	*p* < 0.01	0.41
Making a house and making a space for figures	7 (−3.2)	49 (3.2)	13 (3.2)	17 (−3.2)	10.41	*p* < 0.01	0.35
Task (or exchange) of separation from mother or therapist	24 (2.1)	32 (−2.1)	6 (−2.1)	24 (2.1)	4.49	*p* < 0.05	0.23
Physical excretion and discharge in the play room	10 (−0.2)	46 (0.2)	6 (0.2)	24 (−0.2)	0.06	n.s.	0.03
Direct physical contact with the therapist	15 (1.1)	41 (−1.1)	5 (−1.1)	25 (1.1)	1.12	n.s.	0.11
Client’s entering in the house box or sand tray	17 (1.0)	39 (−1.0)	6 (−1.0)	24 (1.0)	Fisher’s exact test	n.s.	0.11
Emptying something	5 (1.0)	51 (−1.0)	1 (−1.0)	29 (1.0)	Fisher’s exact test	n.s.	0.11
Play looking into the mirror	7 (0.8)	49 (−0.8)	2 (−0.8)	28 (0.8)	Fisher’s exact test	n.s.	0.09
Emergence of the image of height	17 (2.5)	39 (−2.5)	2 (−2.5)	28 (2.5)	6.37	*p* < 0.05	0.27
Conflict with mother	1 (−3.3)	55 (3.3)	7 (3.3)	23 (−3.3)	Fisher’s exact test	p < 0.01	0.35
Expression through images without symbolic development	0 (−2.8)	56 (2.8)	4 (2.8)	26 (−2.8)	Fisher’s exact test	*p* < 0.05	0.30

Note. Adjusted standardized residuals appear in parentheses below group frequencies. The shaded cells indicate statistical significance (defined as *p* < 0.05).

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
