# Peer review of "Gender Differences in Psychological Symptoms and Psychotherapeutic Processes in Japanese Children"

_ijerph, 2020, doi:10.3390/ijerph17239113_

Round 1

Reviewer 1 Report

This study investigates the prevalence and some symptoms of 4 psychological disorders (tics, autism spectrum disorder, trichotillomania and selective mutism) in Japanese children using a psychodynamic approach, through the review of 84 studies published in order to compare the data obtained in children with those obtained in adults. According to the authors, the results show that the prevalence of these disorders is like that of adults. In addition, they analyse certain behaviours (focusing on aggression) in order to observe their appearance in different disorders taking into account gender.

General comment

The approach of this work is interesting, and for that I congratulate the authors. I consider that the scientific analysis of clinical cases is necessary for psychotherapy. However, once reviewed I have found some aspects that should be corrected to provide reliable and objective information to the scientific literature. In addition, English makes it difficult to understand the text correctly.

Firstly, authors name “gender” when they are talking of “sex” (biological sex of participants).

The information of the introduction is not well integrated. Authors describe gender differences in general without relation to the disorders. In addition, the disorders presented are not described from a DSM perspective. On the other hand, it is not justified to focus on these 4 disorders and not others. It should be justified throughout the introduction the reason these disorders are relevant to this study and not others, such as depression, schizophrenia, ... . In addition, the information on the 4 disorders is unbalanced (too much information on some and not enough on others).

Another important aspect is that the references are not updated, only about 5 or 6 articles from the last 5 years are found, being most of them before 2010, which shows that either the authors have not made a correct search or that the topic has not of interest at present. It is very important to update references to give an updated perspective.

However, the most important problem I find in the article is that there is no justification for the objectives and hypotheses. In my opinion, the authors of the manuscript, based on their clinical experience, have hypothesized some aspects but they do not have a scientific justification. All the hypotheses must be justified with scientific literature and those that cannot be justified must be argued. This problem is crucial, given that science requires objectivity and scientific arguments that must be supported by previous studies. In fact, it is well known that there can be cognitive biases that lead to interpreting information in a way that does not correspond to objective reality. In this sense, the fact that there are different “judges” with the same psychotherapeutic orientation, and possibly with similar training, may be implying the existence of these biases. The work would benefit if other therapists from other approaches objectively evaluated the information obtained to see if there is agreement with that obtained by the authors.

Regarding the method of obtaining the data (published clinical cases) I consider that this is also a bias (difficult to solve) since there will be many cases that have not been published and that are also part of the reality that is not in the article. In addition, the articles come from Japanese journals only so there would not be all the studies published on Japanese children. In addition, the criteria for inclusion or exclusion are not presented, nor the databases used, so it is not known how the search was made, which is fundamental for presenting a review. This makes it doubtful that it is a systematic review. Furthermore, an important problem is that the 84 published articles that have been reviewed cannot be identified. It is necessary a table with these 84 publications organized (their references should be in the final list, of course) by disorders where the main aspects are described (authors of the study, year, journal, age, sex, symptoms, time of treatment, ....). This information is essential to know which publications are involved to analyse and check the information presented later.

Regarding the results I can indicate that since there is not a wide representation of some sexes in some disorders it is difficult to talk about sex differences. One girl (or 5, in some cases) is not representative of the girls with a disorder in the world may suffer. Therefore, it should be more cautious in presenting the information and assume in some cases there is not important information (do not provide nothing to literature). This would lead to whether it is necessary to present all 4 disorders or better to select those where there is more sample and useful data can be extracted.

On the other hand, the first point of the results is all interpretation without scientific justification. This part should be omitted or treated in an objective manner. Something similar occurs when tables and statistical analyses are presented. While the analyses are fine, the size of the effect should be included. The p-value, today, is not enough to show consistent differences. In addition, it is possible that the differences found are an artifact of the different n and not the actual presence.

In the “objective” results, the attempt to make a checklist is welcome, but this checklist should be based on scientific literature or at least on clinical experience. In fact, it is slightly biased towards aggression. Moreover, unfortunately, since there is no inter-judge of different psychotherapeutic approach in the creation and interpretation of the checklist, it is difficult to consider that the data shown are not an interpretation of the therapist (a bias).

Finally, the discussion continues in the line that there is a lot of interpretation not based on the scientific literature. In fact, it continues to be very speculative, which is not valid for scientific publications. On the other hand, the fact that more explicit aggressive behaviours are found in boys compared to girls does not necessarily have to be due to the disorders since it is a common trait among boys and girls in healthy populations. 

Furthermore, I miss the utility of the study, given that it is descriptive, but when it is finished it does not give the feeling of providing useful information. What does the article provide to scientific literature besides descriptive data? What would it be useful for a therapist?

And finally, even I am not a native English speaker, the manuscript requires revision of the language since there are expressions that are difficult to understand.

Author Response

Thank you for the thoughtful and constructive feedback you provided regarding our manuscript, “Gender differences in psychological symptoms and psychotherapeutic processes in Japanese children”. We are thankful for the time and energy you expended. Due to the amount of comments we received from the reviewers, we have decided to write our responses in a question-answer format for ease of reading. Our responses to the referees’ comments are as follows:

  1. The approach of this work is interesting, and for that I congratulate the authors. I consider that the scientific analysis of clinical cases is necessary for psychotherapy. However, once reviewed I have found some aspects that should be corrected to provide reliable and objective information to the scientific literature. In addition, English makes it difficult to understand the text correctly.
  2. Firstly, authors name “gender” when they are talking of “sex” (biological sex of participants).
  • This is an important perspective. We revisited the difference between gender and sex. The DSM-5 defines sex and gender differences as follows: "Sex differences are variations attributable to an individual's reproductive organs and XX or XY chromosomal complement. Gender differences are variations that result from biological sex as well as an individual's self-representation that includes the psychological, behavioral, and social consequences of one's perceived gender" (APA, 2013, p. 15). The DSM-5 goes on to state that "the term gender differences is used in DSM-5 because, more commonly, the differences between men and women are a result of both biological sex and individual self-representation" (p. 15). [Sex and Gender in Psychopathology: DSM-5 and Beyond. (2019) Hartung, Cynthia M.; Lefler, Elizabeth K. Psychological Bulletin. 145(4):390-409.]
  • For the above reasons, we decided to adopt the term ‘gender’ in our paper.
  1. The information of the introduction is not well integrated. Authors describe gender differences in general without relation to the disorders. In addition, the disorders presented are not described from a DSM perspective.

- We agree with you and have incorporated this suggestion to our introduction. We significantly modified the composition of our introduction to make it clearer to understand that our hypothesis, derived from previous studies, is that there are gender differences in the four symptoms. We think that these changes resolved the above critique. In addition, the description of symptoms was corrected to follow the latest DSM or ICD description (p.4 Table1, line105-106, line113-116, p.7 line121-124, line133-134). We also refer to the ICD because the DSM does not discuss gender differences for some symptoms.

  1. On the other hand, it is not justified to focus on these 4 disorders and not others. It should be justified throughout the introduction the reason these disorders are relevant to this study and not others, such as depression, schizophrenia,

-We agree with your comment. To this end, we added some brief theoretical and empirical justifications for why these four were selected for analysis (p.2 line50-67). To summarize here, we give the following grounds as justification for picking these four symptoms:

  1. These four disorders are psychiatric diagnoses with strong neurodevelopmental components in childhood [Hansen et al., 2018].
  2. All involve forms of difficulty with direct emotional expression and are tied to disruptive behaviors / tendencies that impair the child’s development.
  3. There are other similar disorders in childhood such as nocturnal enuresis, but we could not find enough number of case studies for analysis.
  4. As you mentioned, there are cases of depression and schizophrenia in children, but the average age of onset tends to be older than our target sample and an insufficient number of cases can be found for inclusion. Schizophrenia usually appears between the late teens and mid-thirties and rarely occurs before adolescence (DSM-5). While depression onset can occur at any age, the greatest likelihood of onset is post-puberty. Given that our manuscript is interested in disorders in childhood (up to age 15), depression is not a primary epidemiological issue for this developmental period (DSM-5).

  1. In addition, the information on the 4 disorders is unbalanced (too much information on some and not enough on others).

- We attempted to improve the balance in the information for each of the disorders, but the number of case studies (e.g. sample size) varied according to each disorder. Thus, there are still some differences in the volume of information for each disorder, particularly for ASD, which had more than double the number of case studies relative to the other disorders.  

  1. Another important aspect is that the references are not updated, only about 5 or 6 articles from the last 5 years are found, being most of them before 2010, which shows that either the authors have not made a correct search or that the topic has not of interest at present. It is very important to update references to give an updated perspective.

- We agree with your comment and updated the references by adding 11 articles, primarily directing the focus of our literature review to contemporary peer-reviewed publications from the last decade.

  1. However, the most important problem I find in the article is that there is no justification for the objectives and hypotheses. In my opinion, the authors of the manuscript, based on their clinical experience, have hypothesized some aspects but they do not have a scientific justification. All the hypotheses must be justified with scientific literature and those that cannot be justified must be argued. This problem is crucial, given that science requires objectivity and scientific arguments that must be supported by previous studies. In fact, it is well known that there can be cognitive biases that lead to interpreting information in a way that does not correspond to objective reality. In this sense, the fact that there are different “judges” with the same psychotherapeutic orientation, and possibly with similar training, may be implying the existence of these biases. The work would benefit if other therapists from other approaches objectively evaluated the information obtained to see if there is agreement with that obtained by the authors.
  • Thank you for providing these insights. You have raised an important point. However, for the current paper, there is insufficient time to collaborate with therapists with different psychotherapeutic orientations.
  • Also, we hope that this paper broadens the understanding of gender differences in psychopathological behaviors in children by contributing an English publication from the perspective of qualitative interpretations in the Japanese cultural context. Based on this, the ratings from the coauthors who are similarly trained is part of the novelty of this paper.
  • Finally, the method we employed which combines expert clinical qualitative evaluations with a clinician/author-derived checklist has been used elsewhere (e.g. Green, E. J., & Gibbs, K. , 2010; Wang, D., & Privitera, A. J., 2019) and this reference was added to p. 5 line 177. We also want to highlight that the researchers first completed their ratings independently of one another before discussing and that the ratings were finalized by combining these independent reviews. By going through this procedure, we think we were able to minimize bias in the ratings as much as possible. 
  1. Regarding the method of obtaining the data (published clinical cases) I consider that this is also a bias (difficult to solve) since there will be many cases that have not been published and that are also part of the reality that is not in the article. In addition, the articles come from Japanese journals only so there would not be all the studies published on Japanese children. In addition, the criteria for inclusion or exclusion are not presented, nor the databases used, so it is not known how the search was made, which is fundamental for presenting a review. This makes it doubtful that it is a systematic review. Furthermore, an important problem is that the 84 published articles that have been reviewed cannot be identified. It is necessary a table with these 84 publications organized (their references should be in the final list, of course) by disorders where the main aspects are described (authors of the study, year, journal, age, sex, symptoms, time of treatment, ....). This information is essential to know which publications are involved to analyze and check the information presented later.
  • Thank you for providing these insights. We have added a list of 84 papers in Table 2. As you pointed out, our research deals only with Japanese cases, which can be said to be the limit of this research. However, there can be cultural differences in therapy and the psychological characteristics of children, so rather than mixing studies from different cultures from the beginning, we first reviewed the studies of a particular culture, with the expectation that future studies can pursue an international perspective. We hope that this study will be the beginning of such research.
  • We agree with you that our research is based on published clinical cases and can be biased by the data. The published cases can be rather successful cases. But because of ethical issues it is getting more and more difficult to have access to unpublished case materials. Furthermore, successful cases are helpful for exploring psychological processes in therapy. We may organize a second research based on our checklist for unpublished cases later. Thank you for this suggestion.
  • We used the Japanese database “CiNii database” for academic articles which is organized by the National Institute of Informatic. We forgot to mention in our first draft, but added in the revised one.

  1. Regarding the results I can indicate that since there is not a wide representation of some sexes in some disorders it is difficult to talk about sex differences. One girl (or 5, in some cases) is not representative of the girls with a disorder in the world may suffer. Therefore, it should be more cautious in presenting the information and assume in some cases there is not important information (do not provide nothing to literature). This would lead to whether it is necessary to present all 4 disorders or better to select those where there is more sample and useful data can be extracted. On the other hand, the first point of the results is all interpretation without scientific justification. This part should be omitted or treated in an objective manner. Something similar occurs when tables and statistical analyses are presented. While the analyses are fine, the size of the effect should be included. The p-value, today, is not enough to show consistent differences. In addition, it is possible that the differences found are an artifact of the different n and not the actual presence.
  • Thank you for your important comment. We agree with your comment and have added effect sizes to Table 3 and Table 4.
  • Concerning your critique for the qualitative results, we found it necessary to first discuss these qualitative analyses to fully illustrate the steps that preceded the quantitative analysis.
  • To improve the reader's ability to objectively review our qualitative results, we have added the number of cases and gender ratio for each characteristic.

  1. In the “objective” results, the attempt to make a checklist is welcome, but this checklist should be based on scientific literature or at least on clinical experience. In fact, it is slightly biased towards aggression. Moreover, unfortunately, since there is no inter-judge of different psychotherapeutic approach in the creation and interpretation of the checklist, it is difficult to consider that the data shown are not an interpretation of the therapist (a bias).
  • We agree with your comment that we should clarify how we made the checklist. We referred to Green, E.J., & Gibbs, K. (2010) as an example of this approach in the field. We added the sentence about this methodology p. 5 line 347-348. Although there may be a variety of behaviors, expressions, or symptoms that occur in therapy for these disorders beyond the checklist we used, the focus of this study was particularly on direct/indirect and externalizing/internalizing expressions, including types of aggression. As such, the checklist we created emphasizes these topics because they are what we are interested in examining in particular.
  1. Finally, the discussion continues in the line that there is a lot of interpretation not based on the scientific literature. In fact, it continues to be very speculative, which is not valid for scientific publications. On the other hand, the fact that more explicit aggressive behaviours are found in boys compared to girls does not necessarily have to be due to the disorders since it is a common trait among boys and girls in healthy populations. 
  • We agree with your comment that the gender differences observed in the current study are not necessarily a phenomenon entirely limited to the clinical samples, and may in fact reflect gender differences that also occur in healthy/typical populations. This would also be consistent with conceptualizing psychopathology and symptoms on a dimension or spectrum, whereby symptoms and disorders are more extreme forms of “typical” behavior. As such, we added the following sentence in p.14 line418-421, “This contrast in clinical samples mirrors general gender differences in healthy, typical samples, in which girls tend to have a tendency to internalize, which may contribute to more differentiated psychological structures (i.e. differentiation of the internal from the external world).”
  1. Furthermore, I miss the utility of the study, given that it is descriptive, but when it is finished it does not give the feeling of providing useful information. What does the article provide to scientific literature besides descriptive data? What would it be useful for a therapist?

- Thank you for suggesting to better highlight the meaning of this paper, especially for a therapist. We mentioned the therapeutic meaning at the end of conclusion of the integration of characteristics of other gender for psychological development.

- The utility of this paper is a broadened perspective of psychopathology from a Japanese psychoanalytic perspective and with a Japanese child sample. For this to be published in English is very novel. Having cases, perspectives, and insights from different cultural contexts is important for contributing to a nuanced and in-depth understanding of the human condition. Qualitative information is beneficial for inspiring thought. Perhaps readers who may or may not be therapists will find this interesting, perhaps some who are therapists will find insight to help them with therapy, but the aim of this paper is not to give explicit therapeutic advice or instructions per se. We leave this to the therapists’ own interpretation and discretion.

  1. And finally, even I am not a native English speaker, the manuscript requires revision of the language since there are expressions that are difficult to understand.
  • We agree with your assessment. We have revised the English and hope that it has become clearer and easier to understand. Two native English-speaking researchers proofread and edited the entire paper’s language.

We look forward to hearing from you regarding our submission. We would be glad to respond to any further questions and comments that you may have.

Reviewer 2 Report

The subject of study as well as the conclusions drawn are of great relevance for equal emotional education from early childhood and for the clinical setting with this type of childhood disorders. The effort of the authors to carry out this double qualitative and quantitative study is noteworthy. However, there are many aspects to improve so that the work acquires the scientific rigor it requires.

Although internalization and externalization is a well-known phenomenon in the phenomenology of disorders. In this case, the qualitative analysis of the data does not make it clear that this is the case, since the therapeutic sessions are not clearly analyzed according to the variables to be studied, internalization versus externalization of behaviors. The therapy carried out is not detailed and whether it was the same for all cases, which may be influencing the results. The variables to observe in the case of the qualitative analysis of the cases are not detailed either. The quantitative analysis is made with a certain predetermination of the qualitative results and without detailing the clear theoretical-scientific basis to support the checklist.

Author Response

Thank you for inviting us to submit a revised draft of our manuscript, “Gender differences in psychological symptoms and psychotherapeutic processes in Japanese children”. We also appreciate the time and effort you have given to provide insightful feedback on our paper. We have incorporated changes that reflect the detailed suggestions you have provided. Due to the amount of comments we received from the reviewers, we have decided to write our responses in a question-answer format for ease of reading. Our responses to your comments are as follows:

  1. Order the information that develops with greater expository clarity. For example, show the gender differences at once and then add the possible explanations for these differences. Find out more about cultural factors and differential emotional education in men and women, review articles such as:Sanchez-Nunez, M., Fernández-Berrocal, P., Montañés, J., & Latorre, J. M. (2008). Does emotional intelligence depend on gender? The socialization of emotional competencies in men and women and its implications.
  • Thank you for your valuable information. We made significant structural/organizational changes to the Introduction with the goal of i) clarifying the hypothesis and ii) rooting the hypothesis and current research objective in the extant literature. We think these major changes has resolved the problem you identified.

  1. (original manuscript p.2 line49)This paragraph should be placed just before the objectives of the study or hypothesis not here, where all the literature has not yet been reviewed.

-  Thank you for your advice. As we wrote above, we think this point has been resolved due to the major revisions of the Introduction.

  1. If this is the same data that appears in the table, it is not necessary to include it again here. On the other hand, if it is better to specify the opposite results that are not reflected in the table as references 21 and 22. If it does not seem that the information is presented in a biased way.

  • Thank you for your suggestion. We edited Table 1 to list the main studies on gender differences in the four disorders’ prevalence. Regarding selective mutism, the DSM does not specifically mention gender differences. However, other peer-reviewed research shows some minor gender differences in the ratio of diagnoses. Therefore, in order to eliminate the bias you pointed out, we have added Reference 21 and 22 to Table 1, while leaving a description of Reference 20 for comparison.

  1. Selective Mutism: In this section, the qualitative analysis does not focus so clearly on the development of internalized and externalized behaviors.

  • You have raised an important question. We thought that we should theoretically distinguish between internalized/ externalized behaviors in the introduction and show some examples with references. We added the explanation of internalized/ externalized behaviors to explicitly link our concept of indirect / direct aggression and how they are conceptually related in terms of the child expressing him or herself (see p.1 line34-38, p.2 line56-59, p.4 line 158-168).
  • We added a brief reference back to this where direct aggression is mentioned in the results. (p.14 line 1419-1420, p.15 line465-473)

  1. (original manuscript p.10 line 302-306) And this behavior could not be considered as direct aggression and therefore externalized behavior?
  2. (original manuscript p.12 line 362) Directly assaulting the mother is not a sign of direct aggression?
  • Thank you for your comment. Their behavior did not directly curse at the mother, but rather the child made his/her mother do something the mother disliked. At first glance, the shared activity can appear to be a source intimately interacting with one other, but in reality, the choice of activity is a way to be mean. Therefore, this is considered indirect rather than direct. In order to show this clearly, we added the explanation of the play concretely: “For example, during sandplay therapy, cohesive themes in play therapy did not develop and play mainly centered around the client’s nuanced and subtle aggression towards his or her mother, such as when a girl forced her mother who was very reluctant to draw a picture to participate in a mutual picture-drawing game with her and her therapist” (p.10 line 1303-1307).

  1. (original manuscript p.9 line262) Qualitative analysis does not present gender differences.

- We added some characteristic behaviors for boys such as hitting and biting the therapist and reaching out to places that are elevated in height.. Furthermore, we included the ratio of boys to girls for each characteristic.

  1. (original manuscript p.10 line311-312, p.11 Table3) Where have these characteristics been extracted? Cite sources.
  2. (original manuscript p.13 line395) The therapy carried out with the children is not explained in detail. Performing the categories of the checklist based on the behaviors observed in the qualitative analysis favors the results found in the quantitative analysis. The origin of the categories developed to make the quantitative analyzes is not specified.
  • Thank you for pointing this out. We understand we should provide more details about how we made the check list. We inserted the phrases at the beginning of quantitative analysis: “To quantitatively evaluate the case studies, the team reviewed and scored all cases against a checklist. Based on the prominent themes from the qualitative analysis and in keeping with the distinction between externalization and internalization of behavior, the research team created a checklist of 23 categories of behaviors and symbolic imagery which were regarded as characteristic and crucial for the therapeutic turning point for each disorder (Table 3). The goal of the checklist was to determine whether the qualitative differences reported above were statistically different.”

  1. (original manuscript p.11 line331-335) Explain more clearly.
  • Thank you for your comment. We have revised the explanation of the tic disorder results.

  1. (original manuscript 13 Table 4) The results tables are not attached for discussion, they are exposed in the results section.

- Thank you for your comment. We changed their location.

  1. There are the boys who have more autism and thus build more enclosed spaces?

- For ASD and tic cases, we argue that the use of enclosed space or passages tends to be used as an outlet for the children’s aggression. On the other hand, girls with selective mutism and trichotillomania tend to be conscious of the inside of the space as a form of internalization.

Because we made substantial change of the conclusion, we could not modify the part you pointed out directly. However, we think that the difference in the meaning of the enclosed space between boys and girls have come to be clearly shown by clarifying the meaning of externalization / internalization.

Thank you again for your comments and the opportunity to resubmit for further consideration. Please contact us if you have any further questions and suggestions.

Round 2

Reviewer 1 Report

Dear authors, 

in my opinion, all my concerns have been fixed. I think it is necessary to publish and share more papers directly related to clinical contexts, so I encourage the authors to perform more investigation, but trying to use a more strict scientific methodology.

Thank you for giving me the opportunity of revise this paper. 

Author Response

Thank you for re-reviewing. We would also like to thank you for acknowledging our corrections. Empirical research on practical data in clinical psychology is difficult in methodology, but we think it is valuable. We would like to continue our research while searching for a valid methodology. Thank you very much.

Reviewer 2 Report

The work has been reviewed with considerable effort and the subject is extremely interesting as it enables advancement in emotional development and mental health.
There are still some issues to be clarified that I highlight in the attached document in relation to your work.

Author Response

Thank you for your additional comments, regarding our revised manuscript. You have provided very important insights. We attach here our revised manuscript, as well as a point-by-point responses to your comments.

  1. on p.2 line 79 (in your attached pdf file) “in that that they are” (highlighted):

Thank you for finding an error. We deleted one that.

  1. On p.4, line 149-155 (in your  attached pdf file) : “Importantly, this study represents an emic approach, as the case studies are about Japanese children… psychological disorders.”

This information is added in future research in this line in the discussion or conclusions section not here.

Thank you for your advice. We moved the phrase “This study represents an emic approach, as the case studies are about Japanese children, reported by Japanese clinicians, and evaluated by a Japanese research team” to the beginning part of the discussion in which we mention the cultural background of this research (p.13, line 3431 in the revised paper). Other sentences in this part have been deleted because they overlapped with already-existing sentences in the discussion about culture and how it relates to the present research.

  1. On page 4 line 157-158 (in your attached pdf file)

This information should be cited before the study objective, as mentioned in the previous review. The study objective is the last thing that is exposed in the review of the scientific literature, just before the Methodology.

Thank you for your suggestion. Because it serves as a summary of the scientific background for the four disorders, we moved Table 1 and the sentence, “A review of the literature showed gender differences in child prevalence for these four disorder categories in epidemiological surveys (see Table 1 for summary)” to just before introducing the research objective (p.3 line 720-721 in the revised paper).

  1. On page 16 line 519 (in your attached pdf file):

Taking into account that in total there are 21 boys and 5 girls. The percentage that presents aggression is lower in girls but 3 of 5 girls would be a high percentage compared to 13 of 21 boys. Explain in detail how the data are interpreted in all the case studies.

Thank you for your comment. As you pointed out, we cannot suggest here that this was more common in boys than girls. So we deleted “and was more common in boys than girls”. The gender difference in direct aggression was only found in the later quantitative analysis between total male and female children, but not for ASD.

  1. On page 16 line 525-537 (in your attached pdf file):

What significance do these results have in relation to the variables that are being studied? Externalization and internalization? And if they are indicated based on what parameters?

Thank you for your comment. These various observations are derived from a qualitative, exploratory analysis of the case studies. These analyses can be interesting for clinicians but there are some results of observations that are not directly connected to our theory and hypothesis. But because most of the qualitative observations have to do with the externalization and internalization, we made this clearer by changing the following sentences:

“Thus, for ASD cases, the qualitative observations made by the clinical team showed variety in expressions. At the same time, they indicate that the externalizing tendency (physical contact with the therapist, physical discharge, and emphasis of height) is dominant in cases of ASD, which progresses toward an internalizing tendency (entering into small spaces and using mirrors) over the course of therapy.”

  1. On page 18 line 625-626 (in your attached pdf file):

This sentence is enlightening, however the qualitative observation was also based on the parameters of DSM? Indicate the sources from which the observation arises and add this information in the limitations of the study that are cited.

Thank you for your inquiry. We made a qualitative analysis of case studies first, using DSM criteria on one hand and our clinical background in sandplay therapy and play therapy on the other hand to form the basis of expressions’ descriptions. However, the categories identified for these characteristic expressions for the disorders can be not only typical/unique for the disorder based on the DSM criteria, but can also be found in other disorders. This is why we made a checklist of categories obtained through the qualitative analysis of case studies and applied to all cases of all 4 disorders to check the difference statistically. The limitation of this procedure is that we used a checklist, which was based on the same cases, which should be mentioned not here, but in the discussion where methodological limitations were listed. We changed the highlighted text and added a text before this accordingly. The limitation is added in the discussion.

“These categories were based partially on the DSM 5 criteria and partially on our clinical back ground in sandplay therapy and play therapy (Roesler, 2019 -> [14]).The objective of the checklist was to determine whether the qualitative differences reported above were statistically different.

In the discussion section page 13 line 3424-3426 (in the revised paper):

 “However, there are several limitations to the current study that should be kept in mind when drawing conclusions. … Second, the checklist created by the research team for this study was not externally created, as it was created based on the case studies that were analyzed.“

  1. On page 23 line 765 (in your attached pdf file): “aggression against mother”

It could be confused with the interpretation of a direct aggression

Thank you for your suggestion. As you pointed out, there was a possibility for confusion, so the name of "aggression against mother" was changed to "conflict with mother" in text and also in the tables.